# Persistent Tachypnoea in Early Infancy: A Clinical Perspective

**DOI:** 10.3390/children10050789

**Published:** 2023-04-27

**Authors:** Samuel Menahem, Arvind Sehgal, Danielle F. Wurzel

**Affiliations:** 1Department of Paediatrics, Monash University, Clayton, VIC 3168, Australia; 2Murdoch Children’s Research Institute, University of Melbourne, Parkville, VIC 3052, Australia; 3Australian Centre for Heart Health, University of Melbourne, Parkville, VIC 3052, Australia; 4Melbourne Children’s Cardiology/Adult Congenital Heart, 53 Kooyong Road Caulfield North, Melbourne, VIC 3161, Australia; 5Monash Newborn, Monash Health, Clayton, VIC 3168, Australia; 6Neonatal Cardiovascular Research, Monash Health, Clayton, VIC 3168, Australia; 7Department of Respiratory Medicine, Royal Children’s Hospital, Parkville, VIC 3052, Australia; 8Allergy and Lung Health Unit, University of Melbourne, Parkville, VIC 3052, Australia

**Keywords:** tachypnoea, newborn, infant, congenital heart abnormalities, congenital pulmonary abnormalities

## Abstract

Tachypnoea in the newborn is common. It may arise from the many causes of the respiratory distress syndrome such as hyaline membrane disease, transient tachypnoea of the newborn, meconium aspiration etc. Congenital heart disease rarely presents with early tachypnoea on day one or two, in contrast to the early presentation of cyanosis, unless there is “pump” (ventricular) failure such as may occur in a cardiomyopathy/myocarditis, or as a result of severe obstruction to either ventricle. Space-occupying lesions within the chest, for example from a diaphragmatic hernia or a congenital cystic adenomatoid malformation, may present with early tachypnoea, as can a metabolic cause resulting in acidosis. The aim of this paper, however, is to focus on infants where the tachypnoea persists or develops beyond the newborn period, at times with minimal signs but occasionally with serious underlying pathology. They include causes that may have originated in the newborn but then persist; for example, arising from pulmonary hypoplasia or polycythemia. Many congenital cardiac abnormalities, particularly those causing left sided obstructive lesions, or those due to an increasing left to right shunt from large communications between the systemic and pulmonary circulations, need be considered. Respiratory causes, for example arising from aspiration, primary ciliary dyskinesia, cystic fibrosis, or interstitial lung disease, may lead to ongoing tachypnoea. Infective causes such as bronchiolitis or infantile wheeze generally are readily recognisable. Finally, there are a few infants who present with persistent tachypnoea over the first few weeks/months of their life who remain well and have normal investigations with the tachypnoea gradually resolving. How should one approach infants with persistent tachypnoea?

## 1. Introduction

Tachypnoea in an infant usually implies significant pathology. It not uncommonly occurs in the newborn. Respiratory distress syndrome (RDS) arising from hyaline membrane disease in the premature infant, transient tachypnoea of the newborn (TTN), meconium aspiration etc., frequently results in tachypnoea of varying degrees [1]. At times there may be the need for respiratory support with an increase in the ambient oxygen. Improvement generally occurs over the next few days resulting in the baby being normally saturated in room air. Rarely tachypnoea on day one or two may be related to a large aortic run-off, away from the lungs (extrapulmonary) which still exhibits a high pulmonary vascular resistance. For example, in such congenital anomalies as a large arteriovenous fistula within the brain, a so-called Vein of Galen fistula [2], and occasionally in the liver [3]. At times cardiac failure may present in the newborn from ventricular or “pump” failure arising from an underlying cardiomyopathy or an acute myocarditis [4,5]. Persistent patency of the ductus arteriosus may delay the onset of tachypnoea in duct dependent systemic circulations such as a tight coarctation of the aorta or critical aortic stenosis, provided there is no early ventricular decompensation [6]. Once the duct starts to close the baby may go on to develop tachypnoea as a result of the failure of the obstructed ventricle, even to the extent of presenting in cardiogenic shock if the duct closes rapidly. In contrast duct dependent pulmonary circulations, such as pulmonary atresia or a severe Fallot’s, tend to present early with cyanosis [6]. Tachypnoea arising from a communication between the systemic and pulmonary circuit may take a while to develop. The newborn’s high pulmonary vascular resistance limits the left to right shunt from, for example, a large ventricular septal defect or patent ductus arteriosus. As a result, tachypneoa in such infants is delayed usually beyond the first one to two weeks of the infant’s life [6,7].

Rarely, a metabolic cause might result in acidosis which may present in the newborn period with tachypnoea following the onset of acidosis with the commencement of oral feeds [8]. Occasionally acidosis may be due to renal failure, for example arising from dysplastic kidneys. Aspiration of meconium, blood or milk, a significant pneumothorax, or an underlying space-occupying lesion such as congenital cystic adenomatoid malformation (CCAM) or a diaphragmatic hernia may lead to early tachypnoea [1]. However, a central diaphragmatic eventration rarely presents in the newborn period but may do so later with tachypnoea and cyanosis [9].

By the end of the first week, but occasionally extending into the second week, tachypnoea from the above causes usually resolves. If persistent, the clinical signs and investigations, which may include a chest X-ray, a detailed cross-sectional echocardiogram and relevant blood tests, generally provide a diagnosis. Yet there are babies who present past the first week of life with tachypnoea which has persisted or developed beyond the newborn period, at times with clear-cut clinical signs but occasionally with minimal or subtle signs or no additional signs apart from the tachypnoea. How should one proceed?

This paper aims to discuss the many causes of tachypnoea beyond the newborn period which vary from those that require early surgical intervention to others that only need ongoing observation.

## 2. Case Report

Baby A was referred at two weeks of age with “effortless tachypnoea without hypoxia”. He was born through an IVF pregnancy, the mother having polycystic ovarian syndrome. She was placed on an immune protocol of daily subcutaneous Clexane and low dose prednisolone for the first eight weeks. She did not smoke during the pregnancy and had her first Pfizer COVID-19 vaccine at 34 weeks when it became available locally. She however developed gestational diabetes, which was managed with diet alone. Baby A was born at term by an emergency Caesarean section with an unreassuring cardiotocograph (CTG). Minimal resuscitation was required. He was managed initially in the nursery for possible sepsis. His cultures proved negative. He briefly developed mild tachypnoea straight after birth but that rapidly resolved. He had a birth weight of 3.87 kg. He was discharged home on day four only to be re-admitted a few days later with tachypnoea noted by the visiting midwife. His respiratory rate ranged between 60–100 per minute. He had normal saturations. His chest was clear and there were no murmurs to hear. His pulses were normal. He was afebrile. His chest X-ray was clear. He was started on antibiotics which were ceased once cultures proved negative. His respiratory rate remained high at about 60–70 per minute. He was discharged home three days later. When referred at two weeks of age he was found to be well, feeding satisfactorily with a good weight gain and again with no abnormal clinical signs apart from his mild “effortless” tachypnoea.

What, then, are the causes that may lead to persistent tachypnoea beyond the newborn period?

### 2.1. Causes Originating within the Neonatal Period

#### 2.1.1. Pulmonary Hypoplasia

Pulmonary hypoplasia may lead to persistent tachypnoea [10]. However, it is important to differentiate it from so-called “dry lung” which tends to be self-limiting [11]. The latter may occur in infants delivered prematurely after prolonged rupture of membranes and are often associated with an immediate onset of pulmonary insufficiency. Their respiratory requirements may be variable, ranging from tachypnoea to needing high-end respiratory support. “Dry lung” is best described as occurring in a premature birth after a prolonged leakage of amniotic fluid (for four days or more), or any other condition leading to fetal loss of lung fluid, followed by severe respiratory distress immediately after birth requiring ventilation with high inflation pressures but with dramatic improvement in respiratory requirements during the first 24–36 h of life [12]. It would be important to exclude other causes such as respiratory distress syndrome or sepsis. “Dry lung” leads to functional hypoplasia of the lung with oxygenation improving dramatically after two to three days of respiratory support. In contrast, infants with pulmonary hypoplasia, whether familial or secondarily related for example to lung compression from large space occupying pulmonary malformations, have a prolonged course and ongoing tachypnoea.

#### 2.1.2. Polycythaemia

Polycythaemia, also called hyperviscosity syndrome, can present with respiratory symptoms which may persist beyond the newborn period. It is defined as a venous hematocrit of >65% or an arterial hematocrit of >63%, while hyperviscosity is defined as a viscosity >14.5 cP at a sheer rate of 11.5/s as measured by a viscometer [13]. The high hematocrit and raised viscosity may lead to apparent cyanosis (presenting with a ruddy bluish appearance), tachypnoea which may persist beyond the newborn period, congestive cardiac failure, and elevated pulmonary vascular resistance. The elevated pulmonary vascular resistance tends to drop after a partial exchange transfusion [14,15]. Chest radiographs may show prominent vascular markings. While the symptoms may be associated with polycythaemia/hyperviscosity, the cause-and-effect relationship remains unclear. Excessively delayed cord clamping, twin-to-twin transfusion and holding the infant below the mother at delivery are common causes of polycythaemia which is also seen in growth retarded or post-mature infants.

#### 2.1.3. Respiratory Distress Syndrome in Infants of Diabetic Mothers (IDM)

Advances in pregnancy management over the decades have resulted in the lowering of the incidence of RDS in IDMs. Delayed pulmonary maturity may occur in IDMs because hyperinsulinemia blocs cortisol induction of lung maturation [16]. In addition, the RDS might be accompanied by an associated hypertrophic cardiomyopathy and/or neonatal polycythaemia, which may contribute to ongoing tachypnoea causing a transient need for respiratory support. A hypertrophic cardiomyopathy may result in significant subaortic stenosis from asymmetric septal hypertrophy, rarely presenting with heart failure and cardiomegaly [17]. Most symptoms resolve by two weeks while the cardiac hypertrophy generally resolves within a few months. Supportive care, transient respiratory support such as nasal continuous positive airway pressure or high flow, and diuretics may occasionally be required.

#### 2.1.4. Transient Tachypnoea of the Newborn (TTN)

TTN also known as “wet lung” generally presents as a mild, self-limited cause of respiratory distress, commonly seen in late preterm or term infants [18]. Chest retraction and cyanosis may also occur requiring oxygen supplementation. Chest radiographs may show fluid in the horizontal fissure.

Lung aeration is important for the transition from intra-uterine to extra-uterine life [19]. If adequate breathing is not initiated following birth, the expected drop in pulmonary vascular resistance will not occur and oxygen cannot reach the blood circulating through the lungs. If the newborn infant does not breathe sufficiently to force liquid from the alveoli into the alveoli tissue, air cannot enter the alveoli resulting in hypoxemia. In extreme prematurity, an added problem is the inability of the premature infant to generate sufficient hydrostatic pressure gradients during inspiration to clear the airways of liquid.

#### 2.1.5. Miscellaneous

Aspiration of meconium or blood may obstruct the airway leading to hypoxemia and at times prolonged tachypnoea.

An underlying structural malformation of the airway may hamper effective breathing and may lead to intermittent cyanosis and/or tachypnoea. These include obstruction of the upper airway by bony cartilage—choanal atresia, the presence of a laryngeal web, pharyngeal or bronchial airway malformations, for example arising from Pierre Robin Sequence or laryngomalacia, respectively [20]. A tracheo-oesophageal fistula with oesophageal atresia presents with choking on secretions or feeds with associated cyanosis. Occasionally, either spontaneously or as a result of an excessive volume of gas delivered by a manual ventilation device, newly born infants might develop a significant air leak, resulting in tachypnoea from a pneumothorax which may be picked up clinically or by a chest radiograph, at times requiring urgent intervention [21].

### 2.2. Cardiac Causes

#### 2.2.1. Duct Dependent Systemic Circulations

Duct dependent systemic circulations arising from such congenital abnormalities as a tight coarctation, severe aortic stenosis or interrupted aortic arch, generally develop tachypnoea once the duct starts to close. That at times may occur beyond the first week of life [6,22]. The clinical signs in such infants are especially helpful and may include the presence of a murmur, differential or absent pulses, differential saturations between the preductal (right hand) and postductal (foot) sites, possible signs of right heart failure with an increase in the liver span and mild peripheral oedema, akin to lymphoedema of the adult [23]. Once suspected, a detailed cross-sectional echocardiogram will clarify the diagnosis. Of note, despite the advances in fetal echocardiography, aortic arch anomalies are frequently missed as often the large ductus in utero tends to overshadow the affected area of hypoplasia or interruption [24,25]. Of importance is that the onset of cardiac failure may lead to the loss of any murmurs and the development of generalized poor pulses.

#### 2.2.2. Large Communications between the Systemic and Pulmonary Circulations

The high pulmonary vascular resistance in the fetus, which falls rapidly with the baby’s first breaths, continues to fall over the first few hours as further aeration of the lung tissue occurs. The expansion of the alveoli results in an increase in the alveolar vascular bed [26]. That is further aided by the expulsion of alveoli fluid which may take a few hours or more to occur. The laying down of new lung tissue will also increase the pulmonary vascular bed over the next few weeks [27], together with the thinning-out of the proximal pulmonary arteries following a drop in the pulmonary artery pressures, further reducing the pulmonary vascular resistance over the next few months of the infant’s life. The systemic resistance remains high so that there is an increasing left to right shunt, for example through a large ventricular septal defect or patent ductus [6]. That will result in congestion of the lungs, with a decrease in its compliance resulting in tachypnoea which will persist as long as there is a large left to right shunt [28]. Other features of right heart failure may also manifest as noted above. Again, the presence of a murmur may well suggest the correct diagnosis, further confirmed by a detailed cross-sectional echocardiogram.

#### 2.2.3. Ventricular or “Pump” Failure

Leaving aside ventricular decompensation as a result of a severe obstructive lesion involving the left or right ventricles, especially relevant if the duct has closed, a cardiomyopathy and/or acute myocarditis may present with tachypnoea, which if severe enough may occur in the newborn period or later, and which tends to persist [4,5]. Clinically, both conditions may be described as “silent” hearts as murmurs are unfrequently heard in the acute phase [29]. Once treated, and with an improvement in the cardiac output, the murmurs of mitral ± tricuspid incompetence may become audible. Often such infants present with intercurrent respiratory infections which are often blamed for the tachypnoea rather than an underlying cardiac cause. A chest X-ray and ECG will be particularly helpful while an echocardiogram ± a cardiac MRI will confirm the diagnosis [29].

A rare congenital heart abnormality, namely an anomalous left coronary artery arising from the pulmonary trunk (ALCAPA), nowadays is readily treatable and must not be misdiagnosed as a cardiomyopathy or myocarditis. That anomaly results in myocardial ischaemia and if severe enough myocardial infarction. Such babies generally present with tachypnoea and cardiac failure when aged four weeks or more as the left to right shunt from the right to left coronary artery increases with the drop in pulmonary vascular resistance [30]. An ECG shows ischaemic changes, and a chest X-ray will confirm appreciable cardiac enlargement with signs of congestion. Again, no murmurs may be audible despite echocardiographic evidence of mitral incompetence that occurs as a result of the left ventricular dilatation [31].

#### 2.2.4. Unobstructed Total Anomalous Pulmonary Venous Drainage (TAPVD)

Unobstructed TAPVD is a particularly difficult fetal diagnosis [22]. In contrast to obstructed TAPVD which presents in the newborn with early cyanosis, if unobstructed the diagnosis is often delayed. As the newborn has a high pulmonary vascular resistance, there is little increase in the left to right shunt so that little if any murmur is heard. If saturations are carried out, they may be mildly reduced to the low 90 s or high 80 s, difficult to pick clinically. Once the pulmonary vascular resistance drops further, an increasing left to right shunt develops. The pulmonary venous return augments the systemic venous return to the right atrium. Provided that there is an adequate atrial defect it allows free passage of both saturated and desaturated blood from the right atrium into the left atrium to maintain the systemic output. However the increasing left to right shunt from the TAPVD may lead to mild tachypnoea, often missed because the increased flow across the pulmonary valve tends to produce only a soft ejection systolic murmur at the upper left sternal edge [32]. These infants often present with recurrent respiratory infections which are thought to be the cause of the baby’s tachypnoea. An ECG here is particularly helpful as it will show evidence of right ventricular hypertrophy often associated with right axis deviation and right atrial enlargement.

#### 2.2.5. Left Sided Obstructive Lesions

Although rare, these abnormalities are mostly congenital but occasionally they are acquired and difficult to diagnose as there is usually little to no murmur, the infant often presenting with tachypnoea. It includes such conditions as a cor triatriatum, where there is a membrane fairly high up in the left atrial cavity which limits the inflow of the pulmonary venous return to the left atrium, supravalvular mitral stenosis or mitral valve stenosis; the latter, for example, arising from a parachute mitral valve where all the chordae attach to a single papillary muscle. All such lesions lead to pulmonary venous congestion, and in turn pulmonary arterial hypertension [33]. A smallish left ventricle which provides reasonable forward flow will have a similar effect with a rise in the left atrial pressures which then is reflected back through the pulmonary veins to the pulmonary arteries causing pulmonary hypertension. Rarely pulmonary vein stenosis, either congenital or acquired as in premature babies that need prolonged respiratory support and who often develop associated bronchopulmonary dysplasia, results in no murmurs except for pulmonary venous obstruction, which tends to lead to reflex pulmonary hypertension [25,34].

All the above lesions tend to cause persistent and at times increasing tachypnoea with often the only additional sign being that of pulmonary hypertension, clinically difficult to recognise. With the rapid heart rate of the infant, there is usually summation of the second heart sounds so that it is not possible to clearly identify pulmonary closure. What may be heard in infants with pulmonary hypertension is a louder summated second heart sound in the pulmonary area as compared to the aortic area (the left upper sternal edge as compared to the right upper left sternal edge) [33]. The ECG will show evidence of right ventricular hypertrophy. Detailed echocardiography aided at times by a cardiac MRI or CT angiogram may provide a definitive diagnosis prior to the consideration of any surgical intervention.

#### 2.2.6. Vascular Rings

While commonly presenting with stridor from tracheal compression, vascular rings may be associated with tachypnoea, a barky cough, and other respiratory symptoms [35]. A double aortic arch in skilled hands may be diagnosed prenatally by detailed echocardiography, pre-warning the neonatal clinician who may well confirm the diagnosis by ultrasound prior to the development of any symptoms [36]. A cardiac MRI and/or CT angiogram however is almost routinely required to diagnose a vascular ring created by a right aortic arch, aberrant left subclavian artery, and a left ligamentum arteriosum often arising from a so-called diverticulum of Kommerell, an outpouching of the thoracic aorta [35]. Stridor, more commonly arising from laryngomalacia tends to improve with time, while that from a vascular ring persists and/or worsens. A barium swallow showing a persistent indentation of the esophagus is a hallmark of a vascular ring prompting one to proceed to further imaging to confirm the diagnosis [35].

While fetal echocardiography has greatly facilitated the diagnosis of serious congenital cardiac abnormalities, particularly in skilled hands, there is considerable difficulty in recognising many of the above lesions. That places the onus on the clinician to consider them in any baby who develops ongoing tachypnoea, with little if any murmur especially with signs of pulmonary hypertension [25,33].

### 2.3. Respiratory Causes

Tachypnoea caused by infective causes such as bronchiolitis, infantile wheeze, pneumonia etc., generally are readily recognised with their acute onset and associated physical signs, at times aided by a chest X-ray. A family history, the time of the year, and viral studies may further aid the diagnosis.

Less common, and at times with limited clinical signs, are such conditions as primary ciliary dyskinesia (PCD), cystic fibrosis, tracheal oesophageal fistula, or a laryngeal cleft.

#### 2.3.1. Primary Ciliary Dyskinesia (PCD)

PCD should be considered in a term infant with tachypnoea, particularly when there is situs inversus, lobular collapse or persistent oxygen requirement [37]. Early diagnosis of PCD is important to reduce respiratory morbidity resulting from chronic and/or recurrent lung infections.

#### 2.3.2. Cystic Fibrosis

Cystic fibrosis may also present with tachypnoea alone. While most cases (>95%) are detected on newborn screening in Australia, a small minority are missed [38]. The presence of meconium ileus or a known family history should prompt referral for a sweat test and/or genetic testing. Some infants present with tachypnoea alone.

#### 2.3.3. Tracheal Oesophageal Fistula, Laryngeal Cleft

Both may present with tachypnoea in early infancy [39,40,41]. A careful history and observation for the presence of a cough with feeding is needed. Chest X-ray findings may be subtle. If aspiration is suspected, as may occur with gastroesophageal reflux in a baby who tends to have frequent vomiting, early speech pathology review and an otolaryngeal assessment are indicated. Such aspiration is common in infants who have bulbar or pseudobulbar palsy, as occurs in cerebral palsy [42].

#### 2.3.4. COVID-19

The SARS-CoV-2 infection in neonates can occasionally lead to tachypnoea. In most instances where there is respiratory morbidity associated with COVID-19, it is due to complications arising from the maternal COVID-19 infection, for example a pre-term birth rate of 8.8% occurs, compared to 5.5% in unaffected mothers, or co-infections with other pathogens, for example Cytomegalovirus and SARS-CoV-2 [43]. Maternal infection after 26 weeks is associated with an increased risk of adverse neonatal respiratory outcomes [44]. A clinical history of SARS-CoV-2 in caregivers and/or positive viral testing will usually confirm whether COVID-19 is the likely cause of the tachypnoea.

The SARS-CoV-2 infection may occur either via a congenital transmission, estimated to be about 2% of cases of maternal infection, or more commonly via a post-natal infection arising from exposure to respiratory droplets from infected caregivers [45]. While most cases of COVID-19 infection in early infancy are mild, a more severe infection may occur if the mother herself develops a severe infection requiring intensive care treatment, or if there is a post-natal infection where viremia rates are high [45].

#### 2.3.5. Other Causes

Interstitial lung diseases and atypical infections e.g., Pneumocystis jirovecji (PJP) are rare causes of tachypnoea [46]. Usually, the respiratory distress is marked and associated with supplemental oxygen requirements. A chest X-ray often shows widespread alveolar or interstitial changes. A CT chest and bronchoscopy are usually indicated. A lung biopsy and/or genetic testing is often needed to confirm the presence/nature of interstitial lung disease [46]. PJP, rare in immunocompetent infants, is diagnosed on PCR of lung washing (broncho-alveolar lavage) [47].

Baby A.

Multiple causes of tachypnoea in early infancy have been described above. How then does one explain Baby A’s tachypnoea which gradually improved, became intermittent and eventually resolved by about the age of two months? Apart from his tachypnoea, there were no other abnormal findings. He fed well, had good weight gains and was developing normally. Vomiting was not a feature unless the milk flow from the bottle was too fast. There was no obvious infection noted. Apart from a brief period of tachypnoea shortly after birth, his neonatal period was uneventful though his mother had gestational diabetes. An earlier chest X-ray was normal as well as a repeat electrocardiogram and echocardiogram, carried out to ensure that nothing had been missed on the first occasion. The baby has remained well since.

While it is possible that Baby A may have had silent aspiration, that seemed unlikely as he was neurologically intact and his chest was clear. In addition, he presented early, had normal oxygen saturations and a normal chest X-ray, all of which would tend to exclude such conditions as neuroendocrine hyperplasia of infancy or pulmonary interstitial glycogenesis [48].

Do infants like Baby A have their central respiratory centre set at a somewhat higher level for reasons that are unclear? Over a period of a couple of months as they mature the centre “resets” with the tachypnoea resolving, and a return of their respiratory rate to normal levels. Certainly, if the tachypnoea had persisted, further investigations would have been warranted. Further study to clarify the basis of the tachypnoea in such infants as Baby A is required and may do away with the need for concern and/or intensive investigations.

## 3. Conclusions

Despite there being serious causes that may lead to persistent tachypnoea in early infancy, at times forewarned by earlier fetal investigations and subsequently confirmed by clinical examination and appropriate investigations, there is a further group of infants who may have subtle clinical signs in addition to their tachypnoea. Again, if a serious cause is considered additional investigations may lead to a diagnosis. Finally, there still remains a few infants with “effortless tachypnoea” who have an increased respiratory rate alone, who remain well, and whose investigations are normal, and where the tachypnoea gradually settles over the next few months. Further study may allow for a better understanding of the cause of their tachypnoea which may allay ongoing concerns and do away with extensive investigations.

## Data Availability

Further clinical details of Baby A available on request.

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
