# Peer review of "Persistent Tachypnoea in Early Infancy: A Clinical Perspective"

_children, 2023, doi:10.3390/children10050789_

Round 1

Reviewer 1 Report

General:

In this article, the authors provide a comprehensive review of the differential diagnosis of persistent tachypnea in infants after an impressive case report. The manuscript is easy to read and the reviewer generally agrees with the content with only a few comments to improve this paper.

Specific comments:

1. Could the authors include an additional sentence to clarify the aim of this article in the abstract and at the end of the Introduction part?

2. In the case report, "CTG" should be spelled out correctly at the first mention.

3. In the second paragraph of "d. TTN" on page 3, the penultimate sentence "Aspiration of meconium..." is less relevant to the pathology of TTN, and seems more appropriate to be within the next "e. Miscellaneous".

4. The reviewer believes that the vascular ring, especially the double aortic arch, is a rare but one of the most important differential diagnoses of persistent respiratory dysfunction in infants that is sometimes difficult to diagnose. Can the authors include this disease in the manuscript?

Reviewer 2 Report

Dear colleagues!

Your post is interesting and informative. However, I have a number of questions that need to be resolved.

1. You should re-read the rules for filing a clinical case. All discussions should be placed in the "Relevance" or "Discussion" section.

2. At the end of the clinical case, you write "What then are the causes that may lead to persistent tachypnoea beyond the newborn

period?" Despite the detailed explanations, the results of the examination, diagnosis and rehabilitation of the patient are not directly presented.

3. The list of references should be drawn up in accordance with the requirements of the publisher and never cite works that have not yet been published (paragraph 18). I am also confused by the large number of references to sources that are more than 10 and 15 years old. What relevance can we talk about in such terms?

Reviewer 3 Report

This is a decent manuscript by Menahem et al., approaching an interesting and equally challenging issue: the persistent tachypnea in early infancy and the associated etiologies.

There is a rather comprehensive review starting from a case report. However, the link between the case of baby A and the subsequent possible causes is not very clearly described. The authors stressed that baby presented no heart murmurs, but in the cardiac congenital disease section they also underline that only imagery is diagnostic as the murmurs may occur or not at all.

Also, the case lacks details concerning the mother: its smoking status, the influence of hormonal medication (keeping in mind that the conception was via IVF).

An additional section concerning the influence of COVID-19 on the respiratory status of the foetus/baby, as its role in the respiratory distress syndrome amongst pregnant women is well- known.

Best regards,

The Reviewer

Round 2

Reviewer 2 Report

Hello. Satisfied with the authors' response.

Reviewer 3 Report

The Authors decently addressed my previous observations.

Best regards, 

The Reviewer